# Safety Profile of Recommended Vaccinations in Adolescents: Data from Surveillance of Adverse Events Following Immunization in Puglia (Italy), 2016–2020

**DOI:** 10.3390/vaccines9111302

**Published:** 2021-11-09

**Authors:** Antonio Di Lorenzo, Francesco Paolo Bianchi, Andrea Martinelli, Sabrina Lattanzio, Antonella Carbonara, Giusy Diella, Silvio Tafuri, Pasquale Stefanizzi

**Affiliations:** Department of Biomedical Science and Human Oncology, Aldo Moro University of Bari, Piazza Giulio Cesare 11, 70124 Bari, Italy; antoniodilorenzo95@gmail.com (A.D.L.); frapabi@gmail.com (F.P.B.); dott.a.martinelli@gmail.com (A.M.); sabrina.lattanzio@uniba.it (S.L.); antonellacarbonara82@gmail.com (A.C.); giusy.diella@uniba.it (G.D.); pasquale.stefanizzi@uniba.it (P.S.)

**Keywords:** AEFIs, causality assessment, vaccination, adolescents

## Abstract

Adolescence is a critical period for immunization, in which the adhesion rate to recommended vaccinations is often lower than desired. Since the safety of new vaccines is one of the most important factors determining vaccination hesitancy, post-marketing surveillance of adverse events following immunization (AEFIs) is recommended by the World Health Organization (WHO) to better understand the safety of these drugs. This report describes AEFIs notified in Puglia (Italy) after recommended vaccinations in adolescents aged 12 to 18 years in 2016–2020 to determine the safety profile of these products in a real-life scenario. This is a retrospective observational study. Data were gathered from the list of AEFIs notified in subjects between 12 and 18 years of age following administration of recommended vaccines in Puglia in 2016–2020. AEFIs were classified according to the WHO’s decisional algorithm, and causality assessment was carried out for serious AEFIs. From 2016 to 2020, 323,627 doses of vaccine were administered to adolescents in Puglia and 50 AEFIs were reported (reporting rate: 15.4 × 100,000 doses). Of these, 17 (34.0%) were classified as serious, and causality assessment identified 13 of them (76.5%) as vaccine related. The most common symptoms were local reactions, fever and neurological symptoms. No deaths were notified. The benefits of immunization in adolescents appear to be greater than the risk of AEFIs for all studied vaccines; in fact, AEFIs occur in less than 0.1‰ of patients and are generally mild and self-limiting.

## 1. Introduction

Adolescence is a particular period of life characterized by changes in intellectual, moral, physical, emotional and psychological development. All of these changes can have a considerable impact on compliance with immunization schedules, as the approach to any preventive method no longer entirely depends on the parents’ and pediatricians’ judgements as in the first years of life. At this age, compliance is the consequence of a more complex process involving the adolescents’ thoughts and opinions; their relationships with their parents, friends and physicians; and the information they receive from the mass media [1].

Notably, adolescence is a phase of life in which the adhesion rate to recommended vaccination is lower than in other phases of life [2]. In particular, the experience and the fear of adverse events following immunization are the main determinants for refusing vaccination [3].

It is therefore necessary to keep following former pediatric subjects during their adolescence, both to encourage healthy behaviors in these soon-to-be adults and to avoid the loss of immunity to various vaccine-preventable diseases (VPDs) in potentially vulnerable patients [4,5].

Immunization strategies in Italy are designed by the Ministry of Health and described in the National Immunization Plan (NIP). Each of the 20 Italian regions must follow the guidelines stated by the NIP, but they are allowed to offer other vaccines to target populations not covered by the NIP itself. Furthermore, since 2012, the Ministry of Health has promoted the “vaccination schedule for life”, an immunization schedule that follows every phase of an individual’s life with the objective of protecting them for the whole duration of their life [6].

As far as healthy adolescents are concerned, the Italian “vaccination schedule for life” contained in National Vaccine Prevention Plan 2017−2019 provides the following:A booster dose of diphtheria–tetanus–acellular pertussis adsorbed-inactivated poliovirus vaccine (DTaP-IPV) at 11 to 12 years of age;A dose of quadrivalent meningococcal conjugated vaccine (MenACYW) at 12 to 18 years of age;Two or three doses of human papillomavirus (HPV) vaccine, according to the vaccine and to the patient’s age (two doses at 9 to 14 years of age with a 6- to 12-month interval or 3 doses for patients 15 to 26 years of age at 0, 1 to 2 and 6 months);One dose a year of influenza vaccine;Two doses of hepatitis A vaccine at 1 to 15 years of age, at 0 and 6 to 12 months in the case of subjects living in areas where hepatitis A virus (HAV) infection is endemic [6].

Puglia, a region in the south-east of Italy with about 4 million inhabitants, is considered an area with endemic HAV infection, mainly due to the common habit of eating raw fish and seafood [7]. Therefore, hepatitis A vaccination is routinely offered in this region as a part of the recommended immunization schedule for children in the second year of life; catch-up is recommended at 12–13 years for subjects not immunized [8].

The 2018 Apulian edition of the “vaccination schedule for life” further recommends completing the immunization schedule with two doses of measles–mumps–rubella vaccine at 0 and 4 to 8 weeks and two doses of varicella vaccine at 0 and 1 month in subjects 12 to 16 years of age that are susceptible to these diseases or did not receive the necessary vaccines during childhood. The two vaccines can be administered either separately or using quadrivalent MMRV vaccine. Furthermore, Puglia’s “vaccination schedule for life” recommends two doses of serotype B meningococcal vaccine in adolescents [8].

Surveillance of adverse events following immunization (AEFIs) in the post-marketing life of new vaccines is therefore recommended by the World Health Organization (WHO) in order to better understand the safety profile and effectiveness of new drugs [9]. Indeed, post-marketing surveillance can lead to the detection of rare sanitary events that were not observed during pre-licensure studies through the revision of reporting rates and the analysis of the vaccine’s safety profile in subgroups that were not represented in its pre-marketing life [10,11]. In order to grant a more ordinate approach to AEFIs surveillance, the WHO has recommended the application of a standardized causality assessment methodology instead of the earlier “emotional” approach, which would threaten to increase vaccine hesitancy [12].

This report describes the AEFIs notified in adolescents in Puglia from 2016 to 2020 following recommended vaccinations. The aim of the study is to design a safety profile of these products in the population of adolescent people and, more importantly, in a real-life scenario, as recommended in WHO guidelines.

## 2. Materials and Methods

### 2.1. Data Sources

This is a retrospective observational study.

Data were gathered from the list of AEFIs notified following B-serotype (Bexsero, Trumenba) and quadrivalent ACW_135_Y-serotype conjugate (Nimenrix, Menveo) meningococcal vaccination, HPV vaccination (Cervarix, Gardasil, Gardasil 9), hepatitis A vaccination (Havrix), trivalent measles–mumps–rubella vaccination (MMR II, Priorix), quadrivalent measles–mumps–rubella–varicella vaccination (Priorix Tetra), monovalent varicella vaccination (Varivax, Varilrix), DTaP (Boostrix) and DTaP-IPV (Polioboostrix) vaccination in subjects between 12 and 18 years of age from January 2016 to December 2020.

The list of AEFIs related to the aforementioned vaccines was then collected from the Italian Drug Authority (AIFA) database. Reporting AEFIs is mandatory for all healthcare workers in Italy; reports must be submitted to the National Pharmacovigilance Network (RNF), an online platform managed by AIFA itself. AEFIs may also be reported by the person experiencing them or by their legal representatives.

The overall number of the vaccines administered per year in Puglia to subjects in the selected age range at time of administration was extrapolated from the regional online immunization database (GIAVA).

### 2.2. Population and Database

The target population was represented by subjects aged between 12 and 18 years who underwent recommended vaccinations from January 2016 to December 2020.

For every subject who experienced one or more AEFIs, a form was filled in including information on date of birth, gender, date of vaccine administration, other vaccines administered at the same time and information about the AEFIs (date of onset and date of computing in RNF, clinical characteristics, case description, duration and treatment, hospitalization or emergency room access, final outcome).

An Excel spreadsheet was used to build the database and perform the required analyses.

### 2.3. Data Analysis

The total reporting rate was calculated as the total number of reported AEFIs/the number of vaccine doses administered in subjects between 12 and 18 years of age, while the annual reporting rate was calculated as the number of AEFIs that occurred in a year/the number of doses administered in the same year in subjects between 12 and 18 years of age.

WHO guidelines were used to classify AEFIs as “serious” or “non-serious”. An AEFI is considered serious if it results in death, is life-threatening, requires in-patient hospitalization or prolongation of existing hospitalization, results in persistent or significant disability/incapacity, is a congenital anomaly/birth defect or requires intervention to prevent permanent impairment or damage. Additionally, in 2016, AIFA published a list of health conditions that must be considered as serious AEFIs when occurring after vaccination. This list is the Italian edition of the EMA important medical events list [13,14].

For serious AEFIs, we retrospectively applied the WHO causality assessment algorithm to classify AEFIs as “consistent causal association”, “inconsistent causal association”, “indeterminate” or “non-classifiable”. In particular, for AEFIs that required hospitalization, we examined data from the patients’ medical records for a better classification [15]. Causality assessment was carried out by two different physicians with expertise in vaccinology, and results were compared; in the case of divergent results, a review of the literature was carried out and a third physician was consulted.

## 3. Results

From January 2016 to December 2020, 323,627 doses of vaccine were administered in Puglia to adolescents aged 12 to 18 years. Further details are provided in Table 1.

During the study period, 50 AEFIs were reported in Puglia (reporting rate RR: 15.4 per 100,000 doses administered). Reporting rates increased over time and peaked in 2019 (Figure 1).

AEFIs reporting rates for each vaccine are provided in Table 2.

The overall male/female ratio in AEFIs was ~0.815 (22 males vs. 27 females). For one patient, gender had not been provided by the reporting subject.

The majority of AEFIs were reported in subjects aged 12 (30, 60.0%), and 8 AEFIs (16.0%) occurred in 13-year-olds, 5 (10.0%) in 14-year-olds, 2 (4.00%) in 15-year-olds, 2 (4.00%) in 16-year-olds and 3 (6.00%) in 17-year-olds. No AEFIs were reported in subjects aged 18.

Table 3 shows the distribution of signs and symptoms described in AEFIs reports and the reporting rate ×100,000 administered doses.

Figure 2 shows the signs and symptoms described in AEFIs reports for each vaccine.

Out of 50 AEFIs, 17 (34.0%) were classified as serious and 32 (64.0%) as non-serious, according to the latest WHO guidelines. For one of the reported AEFIs, data about severity had not been provided. Two out of seventeen serious AEFIs (11.8%) led to the patients’ hospitalization. No cases of death or severe/permanent impairment were reported.

The reporting rate for serious AEFIs was 5.25 × 100,000 doses.

Out of 17 serious AEFIs, 13 (76.5%) were deemed consistently associated with the vaccine’s administration after WHO causality assessment was carried out. In one case (5.88%), no consistent association was found between the serious AEFI and the vaccination, and two AEFIs (11.8%) were described as non-classifiable. The outcome of the remaining serious AEFI’s causality assessment was not provided by the reporting subject.

As for the 13 serious AEFIs classified as vaccine related, 2 of them (15.4%) occurred after vaccination with Bexsero, 2 (15.4%) after vaccination with Nimenrix, 1 (7.69%) after vaccination with Polioboostrix, 7 (53.8%) after vaccination with Trumenba and 1 (7.69%) after vaccination with Varivax.

One of the two Bexsero-related serious AEFIs consisted of lymphadenopathy, while the other reported fever, pain near the injection site, photophobia, bursts of heat and hypertrophy of the tonsils, and the patient was hospitalized. This was the only case in which an AEFI led to the patient’s hospitalization. While the first AEFI was fully resolved by the time data were collected, there is no information about the outcome of the second case. However, since it was reported in 2018 and the vaccination schedule was not affected by the adverse event, it is safe to assume that the patient fully recovered from it.

As for the two serious AEFIs reported for Nimenrix, in the first case, arthralgia, asthenia, reddening near the injection site, fever and lymphadenopathy were reported; the second patient reported headache, fever and vomiting.

The one serious AEFI reported following Polioboostrix vaccination was characterized by loss of consciousness, but no other symptoms were reported.

Trumenba was associated with the highest number of serious AEFIs, with seven serious adverse events having been deemed consistently related to the vaccine. In five out of seven cases (71.4%), fever was reported, and it was the only relevant symptom in two patients (28.6%). A patient (14.3%) also suffered from neurological symptoms, while another reported erythema and pain near the injection site; both symptoms were reported together with fever in a third patient. The only two patients (28.6%) for whom fever was not described suffered from loss of consciousness in one case and generalized urticaria-like rash in the other.

Finally, the Varivax-related serious AEFI was in fact an immunization failure; the patient developed varicella despite having been vaccinated.

The overall reporting rate for vaccine-related serious AEFIs was 4.33 × 100,000 doses.

The outcome of 24 out of 50 AEFIs (48.0%) was the patient’s complete recovery, while for 2 of them (4.00%), only partial resolution occurred. In 11 cases (22.0%) the subject did report at least some improvement of their symptoms, and 6 patients (12.0%) were still suffering from the reported AEFIs. For seven AEFIs, the outcome was not provided by the reporting subject. As already stated, in no cases did the reported AEFIs cause the patient’s death or severe/permanent impairment.

## 4. Discussion

Our study describes data referring to the safety profile of various vaccines recommended for use in the adolescent and currently offered in Puglia in subjects 12 to 18 years of age as part of the region’s routine vaccination schedule. The vaccines were offered actively and free of charge, and 323,627 doses of vaccine were administered to adolescents in Puglia from 2016 to 2020. The significant variance in administrations year by year is likely due to the introduction of new products in the immunization schedule (such as Gardasil9) or the different availability of some vaccines from year to year (this is likely the case for DTaP vaccines and DTaP-IPV). Moreover, some products may have been preferred over others after their market introduction (such as Trumenba over Bexsero).

Data from passive surveillance of AEFIs showed that more than 15 subjects out of 100,000 receiving one of the studied vaccines experienced one or more adverse events. Serious AEFIs were reported in about five cases out of 100,000 administered doses. The most common symptoms were local events of pain, swelling or tenderness, reported in 21 patients; fever and hyperpyrexia, reported in 16 patients; and neurological symptoms, which were observed in 15 cases.

Causality assessment for serious AEFIs led to 13 out of 17 of them being classified as consistently related to the vaccine. None of these adverse events caused either severe or permanent impairment or death.

These data are similar to pre-licensure evidence. The United States Food and Drug Administration (FDA) reported local reactions as common AEFIs (occurring in more than 10% of the subjects injected with the vaccine) in adolescents for Varivax [16], Bexsero [17], Cervarix [18], Gardasil [19], Gardasil 9 [20], Havrix [21] and Trumbenba [22] and noted them as observed AEFIs for MMR-II administration in preclinical studies [23]. The Italian Drug Agency (AIFA) mentioned pain, swelling and redness at the injection site as common adverse events following vaccination with Polioboostrix DTaP-IPV vaccine [24] and Priorix Tetra [25]. Fever was also reported as frequently (~10%) or very frequently (>10%) related to most of these vaccines [16,19,22,23,24,25]. Our results confirm that local events of pain, swelling and redness, as well as fever and hyperpyrexia, are the most commonly observed symptoms in adolescents treated with recommended vaccinations, but the reporting rates are far lower than those observed in pre-licensure studies. For instance, the reporting rate for Trumenba-related AEFIs was 69.1 × 100,000 doses, a far lower value than that displayed in AIFA’s Vaccine Report 2019 [26].

This difference in results may be explained by the different surveillance protocol; pre-licensure evidence is collected through active call, whereas we gathered our data from a passive surveillance network. Passive surveillance determines a higher risk of under-reporting due to the tendency of many Italian patients and healthcare professionals not to report adverse events, especially if mild and self-limiting, a phenomenon that has already been documented by other studies of our research group [15].

A larger study conducted in the United States using data collected from the Vaccine Adverse Event Reporting System (VAERS) and referring to DTaP, meningococcal quadrivalent conjugated, quadrivalent HPV and influenza vaccinations administered from 2000 to 2013 documented a much larger number of AEFIs in adolescent patients, probably due to a greater tendency for reporting AEFIs in a private healthcare service environment, where adverse events may represent an important legal reference for lawsuits. More specifically, 6779 DTaP-related, 7405 meningococcal conjugate vaccine-related, 9652 HPV vaccine-related and 8050 influenza vaccine-related AEFIs were reported in children aged 6 to 17 years. Nevertheless, less than 1% of them was classified as life-threatening, and the study group concluded that the benefits of the vaccinations outweighed the possible risks [27]. Despite the higher numerosity of the target population, this study is flawed by the absence of reporting rates, which were instead included in our analysis.

A South Korean study focused on local reactions following diphtheria–tetanus vaccination in two groups formed by 132 pre-adolescents and 145 adolescents. The latter experienced local adverse events in 68.3% of cases. All observed AEFIs resolved in seven days at most, and no serious events occurred [28]. The higher reporting rate in this study is likely related to the active surveillance protocol chosen by the South Korean study group, which greatly decreased the risk of under-reporting. A similar approach might be employed in future Italian studies in order to improve our understanding of mild AEFIs’ frequency [29].

Lastly, an Australian study carried out from 2007 to 2017 analyzed the safety profile of quadrivalent HPV vaccine in adolescents, collecting data from the Australian Therapeutic Goods Administration. The crude AEFI reporting rate was 39.8 × 100,000 doses, and the most frequent events were headache, syncope and nausea; fever and local reactions were also observed as fairly common symptoms. Serious AEFIs were much less common (7.8% of adverse events), confirming pre-licensure rates [30]. The higher AEFI reporting rate in the Australian study may be explained by the greater attention of both Australian healthcare workers and patients to adverse event surveillance. Moreover, the great difference between the percentage of serious AEFIs in this study and ours may be explained not only by differences in the application of the WHO’s causality assessment algorithm, but also by the tendency of Italian healthcare workers to primarily report serious adverse events, while non-serious ones often remain unnoticed [10,31].

Finally, it is interesting to note that reporting rates dropped significantly during 2020. The decrease in the reporting of AEFIs is likely related to the SARS-CoV-2 pandemic, which both interfered with immunization schedules and raised more urgent matters for all healthcare workers, thus making it difficult to properly report AEFIs.

The main strength of our study is the high numerosity of the reference population, with 323,627 doses of vaccine administered to adolescents over the course of five years, a significantly higher number than the population described in pre-licensure studies. Furthermore, our study described a large number of vaccines with significant differences in their components, while other studies did not focus on as many vaccines. Finally, we studied a population that, despite being a fundamental target of active immunization campaigns, is scarcely or not at all represented in pre-licensure studies, which generally focus on healthy adults.

On the other hand, the main liability of our study is represented by the sources of our data; the RNF is a passive surveillance network, which affects reporting rates and the serious/non-serious adverse events ratio. The missing information for some of the reported AEFIs is a further weakness, lowering the power of the study.

## 5. Conclusions

In conclusion, the risk of AEFIs following recommended vaccinations in adolescent is very low (<0.1% of administered doses), and the risk/benefit ratio for these vaccines is fairly favorable. Furthermore, the few cases of serious adverse events led to neither permanent or severe impairment nor death.

The safety profile of vaccines is currently one of, if not the most important subject of correct medical information, not only because it is often the main debate matter for anti-vaccination groups, but also for its importance as one of the reasons for vaccination skepticism among the general population [32]. One of the most recent examples of failure in communication between the scientific community and the public is the withdrawal of the AstraZeneca ChAdOx-1S anti-SARS-CoV-2 vaccine from the market, which caused a significant distrust toward vaccination in the general population [33,34]. Correct communication should therefore become crucial in the curriculum of healthcare workers in order to build and maintain the trust of the patients towards their physicians.

## Figures and Tables

**Figure 1 vaccines-09-01302-f001:**
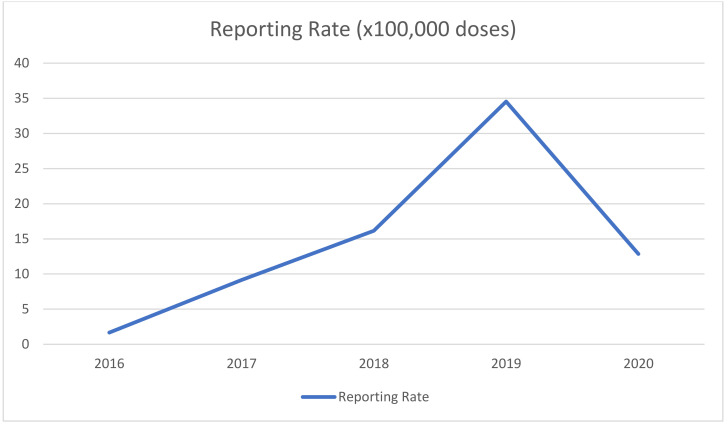
AEFIs reporting rates during 2016–2020.

**Figure 2 vaccines-09-01302-f002:**
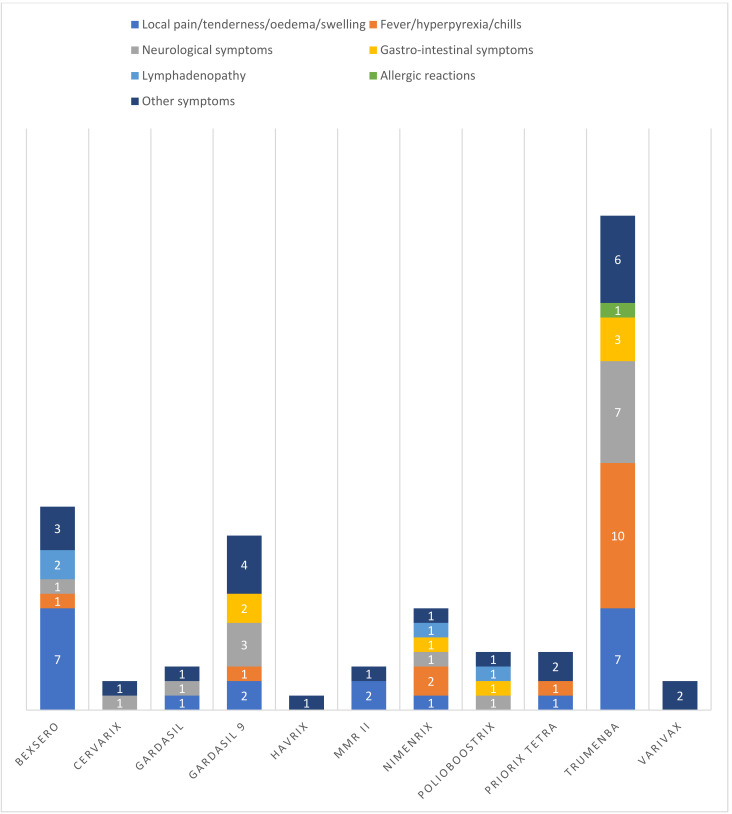
Signs and symptoms described in AEFIs reports, divided by vaccine.

**Table 1 vaccines-09-01302-t001:** Doses of vaccine administered in subjects aged 12–18 years from 2013 to 2020, per type of vaccine and year of administration.

Vaccine	2016	2017	2018	2019	2020	Total
DTaP (Boostrix)	283	945	1962	65	5	3260
DTaP-IPV (Polioboostrix)	13,008	19,882	17,632	12,243	11,339	74,104
HAV (Havrix)	113	119	638	460	243	1573
HPV2 (Cervarix)	16,269	21,634	3140	245	18	41,306
HPV4 (Gardasil)	21,815	1950	511	95	8	24,379
HPV9 (Gardasil 9)	1	476	23,588	26,943	19,311	70,319
Men B (Bexsero)	1249	15,798	20,474	4351	836	42,708
MenACYW (Nimenrix)	1754	623	1277	8911	9566	22,131
MenB (Trumenba)	0	0	620	14,510	12,383	27,513
MMR II (MMR II)	5	15	9	3	2	34
MMR (Priorix)	466	56	14	9	3	548
MMRV (Priorix Tetra)	612	932	1070	45	0	2659
Varicella (Varilrix)	1715	39	9	656	691	3110
Varicella (Varivax)	2611	3074	3338	951	9	9983
Total	59,901	65,543	74,282	69,487	54,414	323,627

**Table 2 vaccines-09-01302-t002:** Reporting rates of AEFIs in subjects aged 12–18 from 2016 to 2020, divided per vaccine.

Vaccine	Number of AEFIs	Reporting Rate (×100,000 Doses)
Bexsero	9	21.1
Cervarix	1	2.42
Gardasil	2	8.20
Gardasil 9	7	9.95
Havrix	1	63.6
MMR II	2	5882.35
Nimenrix	2	9.04
Polioboostrix	3	4.05
Priorix Tetra	2	75.2
Trumenba	19	69.1
Varivax	2	20.0

**Table 3 vaccines-09-01302-t003:** Distribution of signs and symptoms described in AEFIs reports and the reporting rate ×100,000 administered doses.

Signs/Symptoms	N°	% (Out of 50 AEFIs)	Reporting Rate (×100,000 Doses)
Local pain/tenderness/oedema/swelling	21	42.0	6.49
Fever/hyperpyrexia/chills	16	32.0	4.94
Neurological symptoms	15	30.0	4.63
Gastro-intestinal symptoms	7	14.0	2.16
Lymphadenopathy	4	8.00	1.24
Allergic reactions	1	2.0	0.309
Other symptoms	21	42.0	6.49

## Data Availability

The data presented in this study are available on request from the corresponding author. The data are not publicly available due to privacy.

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
