# Peer review of "Safety Profile of Recommended Vaccinations in Adolescents: Data from Surveillance of Adverse Events Following Immunization in Puglia (Italy), 2016–2020"

_vaccines, 2021, doi:10.3390/vaccines9111302_

Round 1
Reviewer 1 Report
It is a very interesting retrospective observational study that will be of the utmost interest for the readers
It would be important to suggest that the reduction in the reported AEFIs during 2020 may be due to the pandemic situation, or any other cause that the authors could identify, and not in the reductions of AEFIs
Author Response
“Finally, it is interesting to notice how reporting rates dropped significantly lower than before during 2020. The decrease in the reporting of AEFIs is likely related to the SARS-CoV-2 pandemic, which both interfered with immunization schedules and raised more urgent matters for all healthcare workers, thus making it difficult to properly report AEFIs.” This line was added to the article. Thank you for your suggestion.
Reviewer 2 Report
This paper aims to analyze the Adverse Events Following Immunization (AEFIs) notified in adolescents in Puglia (a Region in the South-East of Italy) from 2013 to 2020, in order to design these products’ safety profile in a real-life scenario. The paper is interesting, although it should be improved before publication.
English should be checked (i.e "The majority of AEFIs was reported"->WERE).
The introduction section should be improved. First of all, I suggest shortening the first part (lines from "Vaccination" to " life [6]."). The Italian situation is well described, while the paragraphs about the AEFI could be shortened transferring several considerations in the discussion section. Finally, I suggest improving the study's aims.
The Material and Methods section should be improved. I suggest subdividing it into subsections (i.e. Databases, Population, Statistical analysis), inserting several missed information (improve the information about database).
The discussion section should be improved. The authors stated "The overall male/female ratio in AEFIs was ~0.815 (22 males vs. females). For one patient, gender had not been provided by the reporting subject. The majority of AEFIs was reported in subjects aged 12 (30, 60.0%). 8 AEFIs (16.0%) occurred in 13-year-olds, 5 (10.0%) in 14-year-olds, 2 (4.00%) in 15-year-olds, 2 (4.00%) in
16-year-olds and 3 (6.00%) in 17-year-olds. No AEFIs were reported in subjects aged 18. " Despite they introduce data about the sex and age, this data is not indicated in the table. If this information (sex and age) was not reported in the database, the authors should insert it as a limit of the present study in the discussion section.
The discussion section should be improved. I suggest improving the data referred to other countries, discussing them in light of the data of the present study.
Finally, in the conclusion section, in the phrase about the recent situation on the anti-SARS-CoV-2 vaccination, I suggest inserting this missed reference (DOI: 10.3390/diagnostics11060955) that applied the WHO algorithm to two cases of AEFI post-COVID-19 vaccination.
Author Response
Q1. English should be checked (i.e "The majority of AEFIs was reported"->WERE).
A1. The text was reviewed by a native English speaker and several grammatical errors were corrected.
Q2. The introduction section should be improved. First of all, I suggest shortening the first part (lines from "Vaccination" to " life [6].").
A2. Thank you, the introduction section was shortened as asked.
Q3. The Italian situation is well described, while the paragraphs about the AEFI could be shortened transferring several considerations in the discussion section.
A3. We added some considerations in the discussion section.
Q4. Finally, I suggest improving the study's aims.
A4. We tried adding more clarity to our objectives, as suggested.
Q5. The Material and Methods section should be improved. I suggest subdividing it into subsections (i.e. Databases, Population, Statistical analysis), inserting several missed information (improve the information about database).
A5. We revised the section as suggested, dividing it into subsections and adding some missing information.
Q6. The discussion section should be improved. The authors stated "The overall male/female ratio in AEFIs was ~0.815 (22 males vs. females). For one patient, gender had not been provided by the reporting subject. The majority of AEFIs was reported in subjects aged 12 (30, 60.0%). 8 AEFIs (16.0%) occurred in 13-year-olds, 5 (10.0%) in 14-year-olds, 2 (4.00%) in 15-year-olds, 2 (4.00%) in 16-year-olds and 3 (6.00%) in 17-year-olds. No AEFIs were reported in subjects aged 18. " Despite they introduce data about the sex and age, this data is not indicated in the table. If this information (sex and age) was not reported in the database, the authors should insert it as a limit of the present study in the discussion section.
A6. As explained in the Materials & Methods chapter, “For every subject who experienced one or more AEFIs, a form was filled in including information on date of birth, gender, date of vaccine administration”. We did not provided data about gender in the table because it is so big and, if other columns are added, it will bad affect the readability of the table.
Q7. The discussion section should be improved. I suggest improving the data referred to other countries, discussing them in light of the data of the present study.
A7. A discussion of the other studies’ results was added, and references were edited as needed.
Q8. Finally, in the conclusion section, in the phrase about the recent situation on the anti-SARS-CoV-2 vaccination, I suggest inserting this missed reference (DOI: 10.3390/diagnostics11060955) that applied the WHO algorithm to two cases of AEFI post-COVID-19 vaccination.
A8. Added as Citation n. 32.
Reviewer 3 Report
This paper reports post-marketing surveillance of adverse events following immunization (AEFIs) in Puglia (Italy) after administration of a wide range of recommended vaccines in a population of adolescents aged 12 to 18 years between 2016 and 2020. The authors recorded data from a spontaneous reporting system and classified AEFIs for causality using the WHO decisional algorithm. They report that after a total of 323,627 doses administered, there were 50 AEFIs of which 17 were classed as serious, and causally related to the vaccine in 13. Overall, the authors conclude that the benefit of vaccinations in adolescents outweighs the risks.
This is a clear and well presented study that provides valuable information from a population that is targeted by vaccine recommendations, but rarely included in publications. Therefore, the information is important to substantiate the safety of wide-scale adolescent vaccination. At a time when vaccine hesitancy is widespread, and may be on the rise, these data are reassuring and important.
I would like to see some discussion by the authors of the highly variable numbers being vaccinated – For example, the number of doses of DTaP varies from almost 2,000 in 2018 to only 5 in 2020; the numbers getting MMR vaccination (Priorix) plummeted from 466 in 2016 to only 3 in 2020, and for MMRV, from 1070 in 2018 to 0 in 2020. The variations in Gardasil9 figures could be due to the marketing of the vaccine: I see a rise from 1 to 476 to 23,588 over the first years reported – does this correspond to the introduction of Gardasil9 on the market? Since these figures vary widely, and not always in a single-directional trend, I think it would be worthy of some comment.
Otherwise, there are a number of grammatical errors throughout the text. I recommend that the authors have a professional English-speaking medical writer correct their text.
Author Response
Q1. I would like to see some discussion by the authors of the highly variable numbers being vaccinated – For example, the number of doses of DTaP varies from almost 2,000 in 2018 to only 5 in 2020; the numbers getting MMR vaccination (Priorix) plummeted from 466 in 2016 to only 3 in 2020, and for MMRV, from 1070 in 2018 to 0 in 2020. The variations in Gardasil9 figures could be due to the marketing of the vaccine: I see a rise from 1 to 476 to 23,588 over the first years reported – does this correspond to the introduction of Gardasil9 on the market? Since these figures vary widely, and not always in a single-directional trend, I think it would be worthy of some comment.
A1. We added some sentences in the discussion. Thank you for your suggestion.
Q2. Otherwise, there are a number of grammatical errors throughout the text. I recommend that the authors have a professional English-speaking medical writer correct their text.
A2. The text was reviewed and several grammatical errors were corrected.
Round 2
Reviewer 2 Report
Following the reviewers' suggestions, the authors have improved their manuscript. I endorse the publication in its current form.